# Heavy Metal Effects on Biodiversity and Stress Responses of Plants Inhabiting Contaminated Soil in Khulais, Saudi Arabia

**DOI:** 10.3390/biology11020164

**Published:** 2022-01-20

**Authors:** Emad A. Alsherif, Turki M. Al-Shaikh, Hamada AbdElgawad

**Affiliations:** 1Biology Department, College of Science and Arts at Khulis, University of Jeddah, Jeddah 21959, Saudi Arabia; tmalshaikh@uj.edu.sa; 2Integrated Molecular Plant Physiology Research, Department of Biology, University of Antwerp, 2000 Antwerp, Belgium; hamada.abdelgawad@uantwerpen.be; 3Department of Botany and Microbiology, Faculty of Science, Beni-Suef University, Beni-Suef 62511, Egypt

**Keywords:** phytoremediation, amaranthus, heavy metal, sewage, vegetation, bioindicator

## Abstract

**Simple Summary:**

Despite its high organic matter content, sewage sludge contains significant quantities of heavy metals, including those designated as hazardous, such as cadmium, nickel, chromium, mercury, copper, lead, and zinc, which, as a consequence, have a negative impact on living organisms. The current research sought to study the effect of dumping sludge, as one of the sources of pollution with heavy metals, on biodiversity and to assess the bioremediation and stress defense strategies of a tolerant plant species. The obtained results showed that soil pollution by heavy metals has a substantial influence on plant diversity. The selected species, *Amaranthus retroflexus* L., showed a high biological concentration factor (BCF) and low translocation factor (TF) for Cu, As and Ni. The stress defense strategies of *A. retroflexus* grown under complex heavy metals contamination are studied and discussed.

**Abstract:**

Accumulation of heavy metals in soil is becoming an increasingly serious eco-environmental problem. Thus, investigating how plants mitigate heavy metal toxicity is necessary to reduce the associated risks. Here, we aimed to assess the bioremediation and stress defense strategies of tolerant plant species grown under complex heavy metals contamination. To this end, a field study was conducted on the vegetation cover of sites with different soil pollution levels. Forty-two plant species that belong to 38 genera and 21 families were identified. The pollution had a significant impact on plant richness in the polluted sites. Out of several screened plants, *Amaranthus retroflexus* L. was selected because of its high relative density (16.7) and a high frequency (100%) in the most polluted sites. The selected species showed a high biological concentration factor (BCF) and low translocation factor (TF) for Cu, As and Ni. To control the heavy metal-induced oxidative damage, *A. retroflexus* invested in detoxification (metallothionein and phytochelatins, glutathione and glutathione-S-transferase (GST). At the organ level, oxidase damage (H_2_O_2_, lipid and protein peroxidation) was observed, particularly in the roots. To mitigate heavy metal oxidative stress, antioxidant mechanisms (e.g., tocopherols, glutathione, peroxidases, catalase, peroxide dismutase and ASC-GSH cycle) were upregulated, mainly in the roots. Overall, our results suggested the potentiality of *A. retroflexus* as a promising bioremediatory and stress-tolerant plant at the same time; moreover, defense and detoxification mechanisms were uncovered.

## 1. Introduction

One of the major environmental concerns is pollution in many countries, and it significantly challenges global food security. The widespread application of hazardous waste due to the development of industry and agriculture leads to environmental pollution [1]. Each type of pollution has its negative impact on human health, plants, and wildlife, but those that contribute heavy metals to soils and waterways are especially concerning due to their long-term persistence in the environment [2]. Due to overpopulation and the growth of industry, the volume of wastewater is growing daily [3]. Sludge dumping and melting operations all contribute to heavy metal contamination in the environment [4,5]. Although sewage sludge is rich in organic matter, it also contains high levels of heavy metals, including those classified as toxic, such as cadmium, nickel, chromium, mercury, copper, lead and zinc [6]. Moreover, the high concentrations of heavy metals in sewage sludge may induce soil contamination as a consequence, having a negative impact on living organisms. Since soil health is fundamental for food crop production, hazardous metals released into the soil significantly reduce plant quality and yield, which poses a severe threat to humans and animals via their bio-magnification in food chains [7,8]. Heavy metals have a high rate of transfer from soil to feed, which increases their toxicity [7]. Large cities are continuously producing large amounts of effluent water, which must be properly disposed of and managed [9]. Many initiatives, such as phytoremediation, have been attempted to create strategies to remove heavy metals from contaminated soils [4,10]. Phytoremediation has scientific interest, and it has been the focus of several recent studies [4,10]. Phytoremediation using higher plants is an environmentally beneficial method that cleanses toxins from the environment utilizing plants and their related microorganisms [10]. In this regard, phytoremediation using higher plants is frequently used. Moreover, higher plants are eukaryotes; thus, pollution assays can be performed under several environmental conditions, and they are highly reliable and can be used for several years [11].

Pollution drivers can affect individual species, plant groups, and ecosystems on a broader scale. In this regard, the density of perennial and annual species, species diversity, vegetation cover and ecological characteristics associated with the structure of vegetation are among the main indicators of environmental disturbances [12]. Thus, vegetation performs a significant ecological and hygienic role in chemically contaminated soils [13]. For instance, individual species and community spatial dissimilarities may be useful for both measuring pollution effects and identifying potential indicator species [14,15]. Thus, studies on the impact of heavy metal pollution on natural vegetation could contribute to the evolution of species that are both sensitive and tolerant in response to heavy metal pollution [7]. As a result, stress-tolerant plants play a significant part in many environmental processes, incorporating a variety of protective mechanisms [13]. These characteristics may be used to define the condition of the vegetation and to provide better knowledge of floristic composition, demonstrating the development of species that are both sensitive and tolerant in response to pollution exposure and other environmental variables [16,17].

Saudi Arabia’s flora is regarded as the richest in the Arabian Peninsula. The kingdom’s most conspicuous elements of plant life are substantial genetic resources of agriculture and herbal medicines as well as xerophytic vegetation [18]. Saudi flora has a life-form spectrum that is typical of a desert area, with therophytes dominating [19]. Several examples of phytoremediators, such as *Amaranthus retroflexus* L., can be found growing in a variety of soils and textures. It thrives in rich soils and has a high nitrogen demand [20]. It can be found in cultivated fields, gardens, contaminated sites, riverbanks, roadsides, and other open, disturbed environments where annual weeds thrive. It is seldom observed in shady or enclosed locations [21].

We hypothesis that studying the effect of soil contamination with heavy metals on density of perennial and annual species, species diversity and vegetation cover will provide better knowledge of floristic composition as well as identifying stress-tolerant species that can be used as phytoremediators. To this end, we conducted a field study to investigate the disturbance of vegetation cover of sites with different levels of pollution. We also assessed plant phytoremediation, oxidative damage markers and stress-tolerance mechanisms. This research therefore will provide ecological indicators to quantify the impact of soil pollution with sewage sludge on natural vegetation. Moreover, several biochemical markers will be identified to screen for the stress-tolerant plants grown under complex heavy metal contamination. Overall, this research will advance our understanding in the stress mitigation mechanisms and biochemical flexibility of distinct organs of *A. retroflexus*, as well as whether they are connected to organ type.

## 2. Material and Methods

### 2.1. Study Area Description

The study took place near ta sewage dumping lake, which is a man-made lake in the Khulais governorate of Saudi Arabia, in a low-lying floodplain of the mountainous terrain (Figure 1). It has been used as a sewage dumping lake for decades with no cleaning methods in place. It is in the Wadi Khulais watershed (22°8′29.22″ N 39°16′29.12″ E), about 100 m above sea level. Every day, around 100 tanker trucks dump 5.000 m^3^ of wastewater into the sewage dumping lake. It has a hot summer temperature and a warm climate the rest of the year, with little rainfall (Table 1).

### 2.2. Sites Localization

For the vegetation and soil research, five different areas around the sewage sludge lake were selected, each having a 25,000 square meter area. The locations were chosen based on their proximity to the sewage dumping lake, as well as slope, soil type, depth, pH, and area size characteristics. There was no land use at any of the study locations, and there were no disturbances such as livestock, roads, or any other pollution sources. The first site (site 1) is 50 m from the sewage dumping lake, the second is 100 m, the third is 500 m, the fourth is one kilometer (site 4), and the control, the non-polluted area, is five kilometers away (Sc). There was no environmental variation between studied contaminated sites. Furthermore, soils of target sites were all of the same type, and the main drive for the change in vegetating cover is the heavy metal contamination form the sewage area because we found that the closer to the pollution center, the more contaminated the soil with heavy elements, and the further away from the pollution center, the lower the concentration of heavy elements.

### 2.3. Field Surveys of the Study Area

Vegetation surveys were conducted using the quadrats point approach [22]. The sample units were reduced by using points. The research sites’ plant species compositions were obtained by randomly planting one square meter quadrat at ten different places at each site [23]. Species frequency, density, and vegetation cover were all measured. With the use of standard flora reference books [24,25,26,27], plant species within each quadrat were pooled and identified. Life form categories were constructed according to Raunkiaer’s directions [28]. When a taxon has many life forms, the most relevant taxon was picked, with variations in the field life form being disregarded. To avoid the various conceptions of chorological units among authors, the basic technique and language of Zohary [29] for the Saharo-Arabian and Sudanian areas were followed. This resulted in different designations for Saudi Arabia’s two major regions.

### 2.4. Floristic Diversity Analysis

Shannon’s diversity index (S) was used to describe and compare species diversity among the locations studied. The Pielou evenness index (Ep) was calculated [30]. Shannon’s diversity index was calculated:

H’= − Σpi ln pi, where pi = ni/N and ln indicates the natural logarithm.

Pielou evenness index is given as:

Ep = H’/ln S, where S denotes the diversity of species. According to the formula below, the Jaccard similarity coefficient was used to quantify the gradient of diversity changes between the five locations studied.

Cj = (a/(b + c + a)) × 100, where a denotes the total number of species discovered at both locations, b the number of species discovered exclusively at the first site, and c the number of species discovered exclusively at the second site.

Relative density (RD) is a measure of a species’ overall number of individuals in proportion to all other species’ individuals, determined as:

Relative density (RD%) = (individuals’ number of the specific species divided by the total number of all individuals for all recorded species) × 100.

Frequency (F) is the distribution of a species, expressed as a percentage of its total occurrence:

Frequency (F%) = (total number of quadrates studied/number of quadrates where the species occurred) × 100.

### 2.5. Collection of A. retroflexus

At last, five plant samples of *A.*
*retroflexus* were obtained from the rhizosphere at each location, all of which were of the same age and at the same developmental stage. Moreover, the collected *A. retroflexus* plants from different sites were checked to confirmed that they are of same genotype. Roots and shoots were separated, and fresh weights were measured. Collected samples were placed in airtight polyethylene zipper bags, immediately frozen in liquid nitrogen, and then they were transferred to the laboratory. Part of the plant shoot and roots were air-dried at room temperature, weighed to calculated dry weight and stored in appropriate containers until heavy metal analysis including Cd, As, Hg, Al, Fe, Cu, V, Cr, Ni, Co, Pb and Zn, through the digestion of ground plant shoot and roots in HNO_3_/H_2_O (5:1) [31].

### 2.6. Biological Indices

The interaction between plant and mineral, as well as the metal absorption capacity of diverse plant species, was studied using biological indices [32]. By dividing the metal concentration in the plant’s roots by the metal concentration in the soil, the BCF (or biological concentration factor) of metals is calculated. TF refers to the ratio of metal concentration in the shoot to metal concentration in the roots (known as translocation factor).

### 2.7. Heavy Metal and Mineral Content in Soil and Plant Organs

Plant leaves and roots were cleaned with deionized water to remove any apoplastic metal ions. After 150 mg of dry weight of plants and 3 g of soil were digested in HNO_3_/H_2_O (5:1, *v*/*v*), heavy metals and minerals were measured (mass spectrometry, ICP-MS). Following that, standards in 1% (*v*/*v*) HNO_3_ were produced [33,34]. The heavy metal concentration of the soil was measured in g/g DW.

### 2.8. Quantification of Organic Acid

Organic acid (Citric acid) was extracted (Butylated hydroxyanisole in 0.1 percent phosphoric acid). The content was analyzed by using HPLC methods (LaChrom L-7455 diode array) [35]. Ribitol was used as an internal standard.

### 2.9. Photosynthesis- and Photorespiration-Related Parameters

In homogenized plant samples, total chlorophyll a and b, as well as carotenoids, were measured after extraction in acetone [36]. The photorespiration-related essential enzymes (GO, glycolate oxidase and HPR, hydroxy pyruvate reductase) activities were assessed (Feierabend and Beevers [37]). Moreover, the ratio of glycine/serine ratio was known as an indicator of photorespiration [38]. Amino acids such as serine and glycine were measured by using Waters Acquity UPLC-tqd system through separation of amino acids on a BEH amide 2.150 column.

### 2.10. Quantification of Oxidative Damage Markers

H_2_O_2_ levels were measured by using the FOX1 method to monitor the production of Fe^3+-^xylenol orange complex at A_595_ [39]. Lipid peroxidation was measured by using the methods of thiobarbituric acid-malondialdehyde (TBA-MDA) reagent after extraction in 80% ethanol [40]. The different absorbances (440, 532, and 600 nm) were measured, and the content was expressed as nmol. g^1^ fresh weight. Protein carbonyls were measured by Cayman Chemical’s (Ann Arbor, MI, USA) Protein Carbonyl Colorimetric Assay Kit [41].

### 2.11. Quantification of Antioxidant Parameters

To measure the total redox antioxidant capacity (FRAP, Ferric Reducing Antioxidant Power Assay) as well as antioxidant metabolites (e.g., phenolics and flavonoids), samples were extracted by homogenizing them in 80% ethanol by using MagNALyser (Roche, Vilvoorde, Belgium). After centrifugation (14,000× *g*, 4 °C, 25 min), FRAP assay was measured in 0.3 M acetate buffer (pH 3.6) by using tripirydylo-S-triazine reagent. Trolox (0 to 650 µM) as a standard was applied [42]. In ethanol extracts, both polyphenols and flavonoids were detected (Folin-Ciocalteu assay [43]. The modified aluminum chloride technique was used to calculate flavonoid content [44]. ASC and GSH antioxidants were measured by HPLC after separation on HPLC column (Polaris C18-A (100 mm × 4.6 mm, reversed-phase) and after plant samples extraction in meta-phosphoric acid (6%, *w*/*v*). ASC and GSH were detected by a diode array detector (DAD) [34]. Proteins were extracted using potassium-phosphate extraction buffer (50 mM and pH 7.0) for antioxidant enzyme activities. Pyrogallol oxidation at 430 nm [45] in potassium–phosphate reaction buffer was used to determine the peroxidase (POX) [46]. The superoxide dismutase (SOD) enzyme activities were measured by suppression of NBT reduction at 560 nm. Colorimetric analyses of dehydro-ASC reductase (DHAR), monodehydro-ASC reductase (MDHAR) ascorbate peroxidase (APX), and GSH reductase (GR) were performed using the Murshed et al. [47] technique after extraction in 0.05 M MES/KOH. The rate of breakdown of H_2_O_2_ at 240 nm was used to evaluate catalase (CAT) activity [48]. The activity of GSH peroxidase (GPX) was determined by measuring the reduction of NADPH at 340 nm [49]. The Lowry method was used to determine the total soluble protein content [50].

### 2.12. Quantification of Detoxification-Related Parameters

GSH-S-transferase reacted with CDNB (0.5 mM) and GSH (1 mM) in potassium-phosphate buffer (50 mM, pH 7.0). Mozer et al. calculated the activity [51]. According to Diopan et al. [52], the content of metallothionein (MTC) was determined electrochemically by using pulse voltammetry Brdicka reaction. After combining with Ellman’s reagent, the amount of phytochelatins (total thiols-non-protein) was extracted (5 percent sulfosalicylic acid), and spectrophotometry was measured at 412 nm [53].

### 2.13. Statistical Analysis

One-way ANOVA (Tuckey test (*p* < 0.05), SPSS 20.0 software) was applied to estimate the impact of the treatment on roots and shoot of *A. retroflexus* (*n* = 4). The hierarchical cluster was performed by using all parameters (Multi Experiment Viewer (MeV). PCA was performed by using Origin Lab 9.

## 3. Results

### 3.1. Effect of Sewage Pollution on Floristic Composition

Here, 42 plant species were identified. These plant species belong to 38 genera and 21 families (Appendix A). Both Poaceae and Fabaceae (6 species for each) were the major plant families present in the studied control/polluted area, followed by Amaranthaceae and Capparaceae (4 species). Therophytes, Chaemophytes and Phanerophytes were the most frequent life form classes that represented the maximum number of species i.e., 33, 18 and 29%, respectively. Conversely, Geophytes was the lowest class of the life form (1%) (Figure 2). In the most polluted site, *Prosopis Juliflora* (Sw.) DC. had the highest RD (relative density) of 18.2 with a high frequency of 90%. *Amaranthus retroflexus* L. had a high relative density of 16.7 and a high frequency of 100%, followed by *Leptochloa fusca* (L.) Kunth (RD = 9.1, F = 60%). The chorological characteristics of the target species showed that both Arabian Sudanian and cosmopolitan elements showed the highest value (25%), followed by Saharo (20.1%) (Figure 2).

It was observed that the greater the distance between the site and the sewage dumping lake, the greater the vegetation cover (Table 2). The vegetation cover significantly differed between site one (polluted site) and the other four sites (*p* = 0.03). The vegetation cover (Table 2) decreased by 53% in the sewage dumping lake vicinity (site 1), compared to the control (site 5). The species number and diversity measurement of the discovered community are reported (Table 2). The pollution had a significant impact on the plant richness in the area closest to the sewage lake (site 1), where plant richness was decreased by 2.8 times compared with the controlled region (site 5). Furthermore, the change of the characteristic across sites was extremely significant (*p* < 0.05). Table 2 shows that the Shannon–Weiner Index (H′) reflected the ecosystem’s health, with the control site having a higher H′ value of 0.35 than the contaminated sites, which ranged between 0.12 to 0.21. The control site exhibited floristic heterogeneity in contrast to Site 1. Site 1 has more common species than the other sites. There was variability in species composition across Site 1 and the control site, with 23% of common species. It was also found that some sites had fewer common species; for instance, Site 2 had low values of the Jaccard indices. When compared to the control site, the low index reported in the control site was due to differing species occurrences. In the polluted sites (Sites 1–4), *Amaranthus retroflexus*, *Prosopis Juliflora*, *Echinochloa colona*, and *Leptochloa fusca* had high frequency and relative density (RD), suggesting their tolerance for heavy metal buildup by wastewater. *A. retroflexus* has maintained growth in the most polluted site, where moderate decreases in fresh and dry weights (30% and 31%, respectively) were reported. We selected *A. retroflexus*, which exhibited the highest frequencies and relative densities in polluted sites.

### 3.2. Growth Responses to Soil Contamination with Heavy Metals

To evaluate *A. retroflexus* responses to soil contamination, plant biomass (FW and DW), as well as the pigment contents, were measured (Table 2). The results showed that soil contamination significantly induced growth reduction, particularly for the plant roots grown at Site 1 as compared to those grown at Site 5. We measured photosynthetic pigments (Table 2) to investigate the integration of heavy metal accumulation on photosynthetic related parameters and its relationship with the higher growth of *A. retroflexus* plant. The increase was more pronounced (63%) for Cha as compared to Chb and Cha + Chb at Site 5. In contrast, the carotenoids were enhanced in response to the toxicity of heavy metal contamination, compared to those of control plants. Furthermore, this increase was stimulated in plants grown in contaminated soil at Sites 2 and 4.

### 3.3. Metals Accumulation and Uptake by A. retroflexus

To investigate the degree of heavy metal accumulation in the soils of each site, the concentrations of several heavy metal in the rhizosphere soils of *A. retroflexus* plants were measured (Table 3). Three heavy metals, out of 12 detected heavy metals, i.e., Ni, As and Cu showed the highest levels. Depending on the concentrations of these heavy metals, we ranked the contaminated soils into 5 levels, from Site 1 (the highest contamination) to Site 5 (control). Compared to Site 5 (control), Sites 1, 2, 3 and 4 showed gradual increases in the concentrations of Ni, As, and Cu. At the most contaminated site (S4), the content of the three heavy metals represented 87% of the total heavy metals (Table 3). Moreover, Cu was the dominant heavy metal, where its content reached 73% of the total heavy metals. Conversely, soils showed considerable levels of several essential and non-essential minerals, including N, and K, whereas their levels were not significantly affected by heavy metal accumulation in soil (Table 3). The accumulations were measured in both the plant shoots and the root. Ni, As, and Cu levels were sharply increased in both organs of *A. retroflexus* plants and to a greater extent in the roots of plants grown in contaminated soil at site 4 (Table 4). Similar to their level in soil, the highest accumulation was recorded for Cu, more than ten folds, for *A. retroflexus* shoot and roots at site 1 compared to site 5 (Table 4).

Compared to plants grown in control soil (site 5), plants grown in contaminated sites, particularly Site 1, showed increased concentrations of several other heavy metals (Cd, Cr, and Zn). Moreover, mineral nutrition was disturbed due to the competition with heavy metals. Thus, the concentrations of essential plant nutrients (e.g., N and K) in both shoot and roots of *A. retroflexus* plants were evaluated in the present study to determine the state of plant nutrition (Table 4). Heavy metal accumulation reduced mineral uptake by both organs, and this effect was strengthened in the case of roots (Table 4). Mg and Fe uptake were significantly reduced by 72% and 94% in the *A. retroflexus* shoots and roots grown in the most contaminated soil, respectively (Table 3).

The results indicated high uptake (BCF), low translocation from roots to shoot (TF) and high content in both roots and shoot for Cu, As and Ni (Table 5). The biological concentration factor (BCF) ranged between 18.6 and 40.4 for Cu, 31–39 for As, 62–95.6 for Ni, 46–101 for Pb and 53–101 for Co. The translocation factor (TF) of metals in *A. retroflexus* ranged between 0.09 and 0.33 for Pb, 0.15–0.27 for Zn, 0.09–0.33 for both Cu and As, and 0.83–3.15 for Co. In the case of *A. retroflexus* growing in control site, the BCF ranged between 21.4 and 62 for the detected heavy metals, while the TF of metals ranged between 0.08 and 0.70.

### 3.4. ROS Production and Oxidative Damage

To understand the downstream effects of heavy metal contamination on ROS levels and production, we investigated their effects on photorespiration, the main source of ROS. In this context, the photorespiration-related enzymes glycolate oxidase (GOX), hydroxy-pyruvate reductase (HPR) and indicator (Gly/Ser ratio), as well as H_2_O_2_ accumulation in response to the toxicity of heavy metal accumulation, were investigated. Plants grown in contaminated soil at Sites 2–4 showed significant increases in H_2_O_2_ by 6%, 9.5%, and 22% in shoot tissues and 60%, 93%, and 130% in roots tissues of *Amaranthus* plants, respectively, as compared to their corresponding control plants (Figure 3). Consistent with the heavy metals-induced H_2_O_2_ accumulation, a significant increase was observed in the photorespiratory indicator Gly/Ser ratio and the GOX, HPR, and CAT activities, mainly in *A. retroflexus* plants grown in the highest contaminated soil of site 4. In more detail, exposing plants to heavy metal stress at Sites 2–4 increased GOX by 28%, 69%, and 158%, respectively, Gly/Ser ratio by 14%, 81%, and 82%, and HPR by 126%, 274%, and 153% (Figure 3).

### 3.5. Heavy Metals Induced More Oxidative Damages in A. retroflexus Roots

To investigate if heavy metal induced mitigating oxidative responses in *A. retroflexus* and if there were organ-specific responses, we quantified heavy metal-induced malondialdehyde (MDA, used as a marker of ROS induced peroxidation of the lipid) and protein oxidation (PO) (Figure 3). High heavy metal accumulation in Site 3 and/or Site 4 had a significant effect on oxidative stress markers in both organs of *A. retroflexus* plants. Heavy metal contamination resulted in significant increases in MDA by 10% and 77%, and PO by 78% and 60% in shoot and roots tissues, respectively, as compared to their corresponding control plants (Figure 3).

### 3.6. A. retroflexus Antioxidant Defense System

Plant metabolism involves numerous oxidative reactions essential for cell viability under heavy metal toxicity, where plants use different antioxidant arsenals to mitigate oxidative stress induced by heavy metals. Out of the contaminated soil of the four sites, soil from Site 4 showed the highest impact on the antioxidant defense system. First, we determined the total antioxidant capacity (FRAP) as well as the level of antioxidant metabolites and activity of antioxidant enzymes. Our results revealed that plants grown in soil contaminated with heavy metals recorded significant increases in FRAP by 60% and 81% in the shoot and roots of *A. retroflexus* grown in Site 4, respectively. We also observed similar patterns for both polyphenol and flavonoids contents, which are the main contributors to FRAP changes, where heavy metal contamination increased them by 27% and 139%, respectively, in *A. retroflexus* roots tissue (Figure 3). Changes in the lipid antioxidant (tocopherols) and the ascorbate–glutathione (ASC-GSH) cycle related metabolites were measured (Figure 3). The roots of Amaranthus plants exposed to heavy metal stress exhibited significant increases in ASC (380%), while shoots and roots showed an increase in GSH (22% and 17%, respectively) and tocopherols (158% and 223%, respectively) levels (Figure 3). Further, heavy metal stress did not significantly affect ASC/TASC or GSH/TGSH ratios in *A. retroflexus* shoots but significantly affected these ratios by 31%, 59%, 58%, and 68% in roots, respectively, relative to the control plants (Figure 3).

Figure 4 depicts changes in the direct ROS scavenging enzymes activity including POX, SOD, and CAT, as well as the ASC-GSH cycle enzymes, in *A. retroflexus* roots and shoots exposed to heavy metals. Activities were differentially increased in the shoots and roots of *A. retroflexus* plants exposed to heavy metal treatments. Mainly in roots tissues, POX, SOD and CAT showed remarkable increases in their activities (by 258, 585% and 85%, respectively) under heavy metal contamination in Site 3 compared to their values in *A. retroflexus* plants grown in the control site. When heavy metals reached the highest levels at Site 4, such increases in POX, SOD, and CAT in shoots and roots increased by 330, 284, 152, 77, 105, and 136%, respectively. Furthermore, heavy metal stress-induced the activity of ASC/GSH recycling enzymes (APX, DHAR, MDHAR, GR, GPX) in both plant organs when compared to the corresponding controls (Figure 4). The highest heavy metal accumulation significantly decreased the level of ASC metabolizing enzymes (APX, DHAR, and MDHAR) and GSH metabolizing enzymes (GR, GPX) in both Amaranthus plants’ organs, but these activities did not significantly enhance in both *A. retroflexus* organs compared to the corresponding as-alone treatment.

### 3.7. Heavy Metal Detoxification Was More Pronounced in A. retroflexus Roots

We measured metallothioneins (metal binding proteins that regulate metal sequestration, MTC) and phytochelatins (gsh oligomers that sequester metals to the vacuole), total gsh, and glutathione-S-transferase (GST), which regulate conjugation of GSH-metal) to better understand heavy metal detoxification mechanisms in both plant organs [36]. Heavy metal contamination increased the levels of total glutathione (Tgsh), and MTC and activity of GST in *A. retroflexus* roots and shoot, but the level of phytochelatins was only increased in roots, compared to the corresponding control (Figure 3). However, heavy metal accumulation in Sites 1 and 2 had no impact on levels of phytochelatins, Tgsh, and MTC, and the activity of GST as compared to the soil of controlled Site 0 (Figure 3). In addition, as compared to the shoot tissue of *A. retroflexus* plants, roots tissue shows higher levels of phytochelatins, Tgsh, MTC, and GST activity under heavy metal accumulation at sites 3 and 4.

### 3.8. Organ and Site-Specific Responses Are Supported by PCA Analysis

To understand the specific responses of the roots and shoots of *A. retroflexus* plant species to heavy metal stress, we performed a principal component analysis (PCA) with an oxidative stress, antioxidant, and detoxification data set. The PCA embodied uniform metabolic/enzyme parameters along the first two dimensions (PC1 and PC2) that declared 48% and 17% of all data variability, respectively (Figure 5).

PC1 separated the measured parameters based on sites’ induced oxidative and defensive responses (57% of all data variables), whereas the organ-specific responses were resolved along PC2 (13% of all data variables). For control and low-stressed *A. retroflexus* plants, PC1 depicted that low heavy metal accumulation induced ASC level mainly in the roots of Amaranthus plants grown in Site 1 and Site 5, and this effect was less pronounced in their shoots. PC1 clustered data of stress-related parameters such as photorespirations, antioxidant, detoxification, and oxidative stress markers in the roots of highly stressed plants grown in Sites 2–4, as well as stressed *A. retroflexus* plants grown in Site 4. PC2 showed organ specification in response to both control and heavy metal stress (Figure 5).

## 4. Discussion

### 4.1. The Species of the Study Area’s Life Form and Chorology

The identified species life-form spectrum in this study is typical of an arid desert region, where thyrophytes predominate, which is consistent with a previous study of the entire Khulais region [18]. These findings support the theory that arid overgrazing, climates and trampling, all of which are common on grasslands, increase therophytes % by introducing and spreading weedy grasses and forbs of related life forms [54,55]. The studied area belongs to Nubo-Sindian Province, a part of the Sudanian Area [28] or to the Nubo-Sindian local center of endemism, a part of Saharo-Sindian geographic zone [56]. It also stretches along the Red Sea coast north of Makkah and the Arabian Gulf coast in Saudi Arabia. Plant species in the study area are adapted to both aridity and extremely high temperatures in this difficult environment; thus, the Saharao-Arabian elements have the largest number.

### 4.2. Effect of Heavy Metal Contamination on Plant Cover and Biodiversity

Pollution is, in reality, a key element determining environmental variability. Pollution’s impact on plants is no longer limited to its morphological, biochemical, and physiological characteristics. Previously, it was recorded that the distribution patterns, associative groupings, and vegetation cover, conversely, are comprehensive indicators for evaluating contaminants [57]. Temperature, water availability, and environmental heterogeneity were all utilized to link species richness variability. As a result, it may help to reconfigure species richness [58]. The current results showed an increase in species richness at distant sites and a decrease at the closest site to the sewage lake (Table 2), which imply a healthier community in remote sites and land degradation in the vicinity of the sewage lake. Similar studies by Blanár et al. [59] and Bayouli et al. [15] demonstrated that species richness, as well as species diversity, had altered along a pollution gradient. Boutin and Carpenter [60] reported a significant difference in species richness and composition between heavy metals-affected sites and controlled (less contaminated) regions, which is consistent with the current findings.

The plant cover has long been employed as an indication of ecosystem health in addition to a climate change and pollution tracker. Several researchers have found that as environmental changes occur on a spatiotemporal scale, vegetation cover decreases or increases [61,62,63]. which implies a healthier community in remote sites and land degradation in the vicinity of the sewage lake. Moving away from Site 1 resulted in higher percentages of plant cover, showing that Site 1′s cover was deteriorating. In addition, a decline in species counts was observed based on a contamination gradient. Furthermore, plant cover is critical in determining the causes of both environmental and biological change [63]. The index of Shannon was used to measure the diversity. Remembering that the index of the diversity spans the index of diversity at the control site, it was higher than at other sites, ranging from highest when all species existed at about the same relative frequency at a place to lowest only when only one species was present [64]. The last behavior is probably attributable to the appearance of new species inhabiting the control area. These species may be more vulnerable to contamination in the soil. (e.g., *Astragalus vogelii* and *Ochradenus baccatus*). Species that are present in remote sites could be susceptible to pollutants from sewage lake. Therefore, their presence in the control site increases the diversity of the site. The Jaccard Index reveals significant variations in plant species richness between the control site and Site 1. This would validate the theoretical distribution of tolerant and sensitive species.

### 4.3. Tolerant and Sensitive Species in the Study Area

The vegetation of polluted areas is a valuable mine of tolerant species. In our study area, some tolerant species showed their ability to colonize areas close to the sewage, such as *A. retroflexus*, *Echinochloa colona* and *Prosopis Juliflora*. The latter also exhibited high tolerance to heavy metal toxicity, and they were distributed along four studied polluted sites. The other species including *Aizoon canariense*, *Aerva javanica* and *Leptadenia pyrotechnica* appeared to be saved in the control site, indicating high sensitivity to the toxicity of heavy metals. The tolerant species could also be pollutants bioremediators [65]. Out of the identified tolerant species, we selected *A. retroflexus* because it showed the highest frequencies and relative densities at contaminated sites. It can survive the toxicity of several heavy metals accumulations i.e., Ni, As and Cu. We also investigated the growth responses and biochemical mechanisms underpinning stress tolerance of heavy metals in *A. retroflexus* plants. Bioremediation is an efficient process can be carried out by higher plants that bind and remediate soil pollutants [6].

### 4.4. Heavy Metal Uptake and Detoxification in A. retroflexus

In this study, we evaluated *A. retroflexus* tolerance in terms of biomass, distribution and high relative frequency and density and its potentiality to uptake, translocate and accumulate heavy metal more in roots tissues. To elucidate whether the variation in tolerance of *A. retroflexus* is related to its heavy metal (mainly Ni, As, and Cu) uptake, translocation, and accumulation, we calculated the heavy metal content in shoot and root, in addition to BCF and TF factors. The BCF of metals in the roots portion of a plant indicated the bioremediation potential of *A. retroflexus*. Moreover, there was low translocation of the detected metals to shoot tissues, which shows phytostabilization suitability.

To avoid injuries induced by heavy metal accumulation in *A. retroflexus* tissues, several detoxification mechanisms have been developed [66]. To reduce heavy metal uptake, plants increased polyphenols and organic acid (e.g., citric acid) exudation into the soil (Table 3). In this regard, increased organic acids exudation by roots can act as electron carriers or as ligands for metal binding to reduce their accumulation [67]. Compared to plants grown in control soil, heavy metal accumulation increased phenol release in the rhizosphere soil of *A. retroflexus* plants, and the release of citrate was observed mainly in the case of plants grown in the most contaminated soil, compared to their corresponding controls. Plants detoxify heavy metals through a various mechanism such as induction of metallothioneins (MTC), phytochelatins (PHCHEL), and GST [15,67,68]. Here, we found a marked induction in the detoxification metabolites, i.e., chelators and phytochelatins (PCs), that are utilized in chelation and sequestration of Cd, Cu and As metals in the vacuole [69]. Similar to our study, the synthesis of PCs is stimulated by heavy metals such as Cd, Cu and As exposure and Cd is often sequestered in vacuoles as Cd-phytochelatins complex [70]. Chelation of heavy metal ions with PCs to nontoxic chelates in vacuoles is one of the key detoxification mechanisms employed by the heavy metal tolerant plant [71]. Moreover, the high ability of *A. retroflexus* accumulation was concomitant with induced detoxification and sequestration mechanisms as indicated by high GST activity and accumulation of GSH. In this context, several studies documented the effectiveness of reduced glutathione (GSH) in heavy metal stress tolerance, where it acts as a metal’s ligand in the cytosol [72].

### 4.5. A. retroflexus Maintained High Growth under Heavy Metal Stress

The threshold for toxicity of the heavy metal varies significantly between plant species, ecotypes, and cultivars. Compared to other plant species grown in the contaminated soil, *A. retroflexus* showed enhanced tolerance and growth improvement under heavy metal stress. In addition, *A. retroflexus* distribution analyses show a significant abundance, revealing a less sensitive behavior under stress induced by heavy metals.

Heavy metal-induced growth reduction can be attributed to the fact that high levels of heavy metals impair photosynthesis and disturb redox status, leading to accumulation of ROS [73,74]. Under heavy metal stress, *A. retroflexus* plants showed less reduction in chlorophyll pigment, indicating maintained photosynthesis efficiency. The increase can be explained by an increase in carotenoids, a powerful antioxidant and photosynthetic protective pigment [75]. The observed decrease in dry biomass can be explained by the heavy metals’ competition with nutrients, thus limiting nutrients uptake and inhibiting the key metabolic enzymes. Consistently, soybean and rice shoot growth were inhibited by Cu, and Zn caused Fe deficiency symptoms and rice was the more sensitive species [76]. Similarly, tomato plants under Ni treatment decreased plant organs biomass, mainly due to disturbance of essential element absorption [77]. We also found less competition between heavy metals and mineral uptake, where *A. retroflexus* plants showed the ability to maintain high levels of essential minerals (e.g., Mg, K and Ca).

### 4.6. A. retroflexus Showed High Redox Balance

ROS generation is one of the main responses to heavy metal oxidative stress [78]. A moderate accumulation of H_2_O_2_ within *A. retroflexus* could play a signaling role, as it has a relatively long lifespan and high ability to cross plant membranes [75]. ROS signaling in stressed plants increases their resistance to heavy metal stress [79]. In this context, it was hypothesized that a slight accumulation of H_2_O_2_ in plant roots could be a protective mechanism through increased roots cell lignification that could act as an apoplastic heavy metal trap [80,81]. Conversely, heavy metal altered biomass accumulation in roots- and shoot-induced overaccumulation of ROS was accompanied by oxidative stress induction as indicated by increased levels of electrolyte leakage, H_2_O_2_, NADP-oxidase, as a source of superoxide anions, and lipid peroxidation. For instance, AbdElgawad et al. [34] have shown that Zn, Cu, Pb and Ni exposure induced oxidative damages in leaves as indicated by high peroxidation of plant lipid. In our study, the accumulation of H_2_O_2_ under As stress could be ascribed to the heavy metal-induced photorespiration related parameters as photorespiration is one of the key H_2_O_2_ generating mechanisms [81].

Although *A. retroflexus* plants accumulated high levels of heavy metals, they showed fewer heavy metal toxicity symptoms, including oxidative stress. We suggested *A. retroflexus* keeps ROS-induced oxidative stress at low levels by increasing the antioxidant system. To scavenge H_2_O_2_ and to minimize the oxidative damage caused by environmental stress, antioxidant biosynthesis is induced as an adaptive response [33,34]. In this regard, Ni, Cd, and Cu accumulation have induced an increase in non-enzymatic and enzymatic antioxidants in plants’ organs and, to a greater extent, in roots. The increase in the activity of these antioxidant enzymes under metal stress was observed in different plant species [82]. For instance, plants showed high activity of APX and SOD under Ni and Cd treatment conditions [83,84]. Heavy metals also boosted the activities of all ASC/GSH biosynthetic enzymes. This improvement is embodied in improving the levels of ASC and GSH in grasses grown in polluted soils [33].

## 5. Conclusions

Soil pollution with heavy metals not only led to a change in the community structure and in the frequency of species, but also to the disappearance of some species. Therefore, the novelty of this work is that we introduced a promising heavy bio-accumulator and stress-tolerant plant at the same time, and we uncovered its defense and detoxification mechanisms. All these data can be used as research objects for molecular intelligence breeding and for providing a basis for the creation of heavy metal-tolerant crops. However, further investigations are needed to identify the molecular mechanisms underlying heavy metal stress responses in roots and shoot organs in plants of *A. retroflexus*.

### Future Perspectives and Recommendations

We recommend studying other plants that are registered in the study area, such as *Echinochloa colona*, *Prosopis Juliflora* and *Leptochloa fusca* as phytoremediators.

## Figures and Tables

**Figure 1 biology-11-00164-f001:**
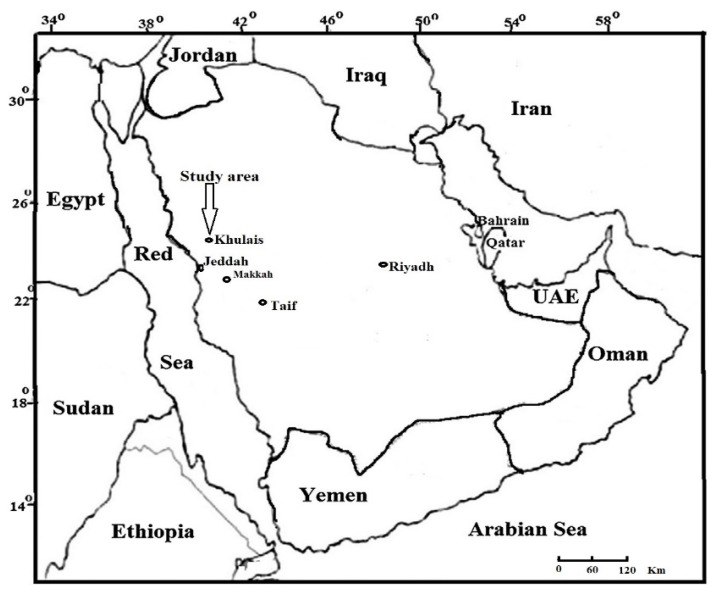
Map showing the study location. The arrow indicates the study area.

**Figure 2 biology-11-00164-f002:**
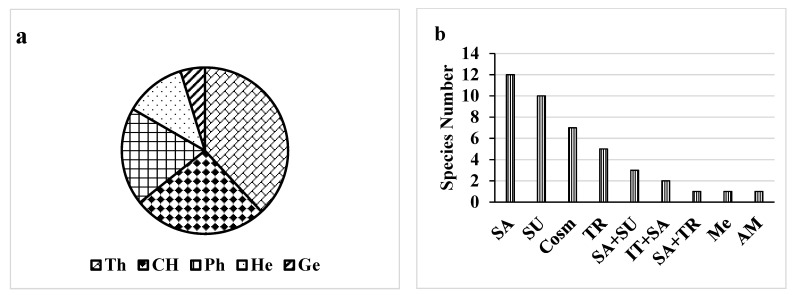
Life form (**a**) and chorology (**b**) of the recorded species. The life forms are: Ph, phanerophytes; Ch, chamaephytes; G, geophytes; He, hemi-cryptophytes; Th, therophytes. The chorotypes are: Cosm, cosmopolitan AM, American; IT, Irano-Turanian; Me, Mediterranean; SA, Saharo-Sindian; SU, Sudano-Zambezian; TR, Tropical.

**Figure 3 biology-11-00164-f003:**
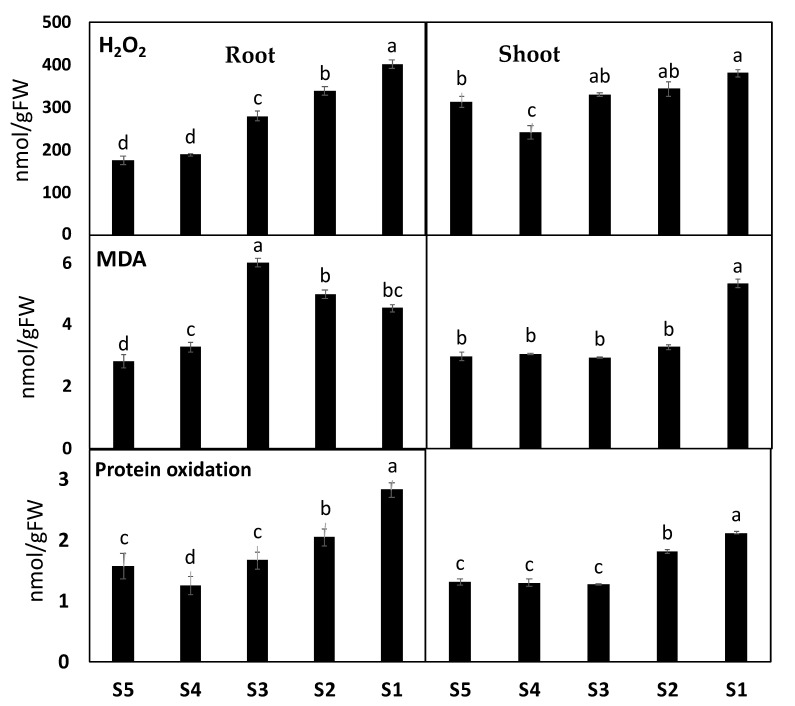
The impact of soil pollution on oxidative stress markers in shoot and roots of *Amaranthus retroflexus*. The changes in hydrogen peroxide (H_2_O_2_), malondialdehyde (MDA) and protein oxidation (PO) in *A. retroflexus* grown in control site (Site 5) and contaminated sites (Site 1–4). Data are mean values ± SD (*n* = 4). Different letters indicate statistically significant difference between means of the same plant species at significance level of at least *p* ≤ 0.05.

**Figure 4 biology-11-00164-f004:**
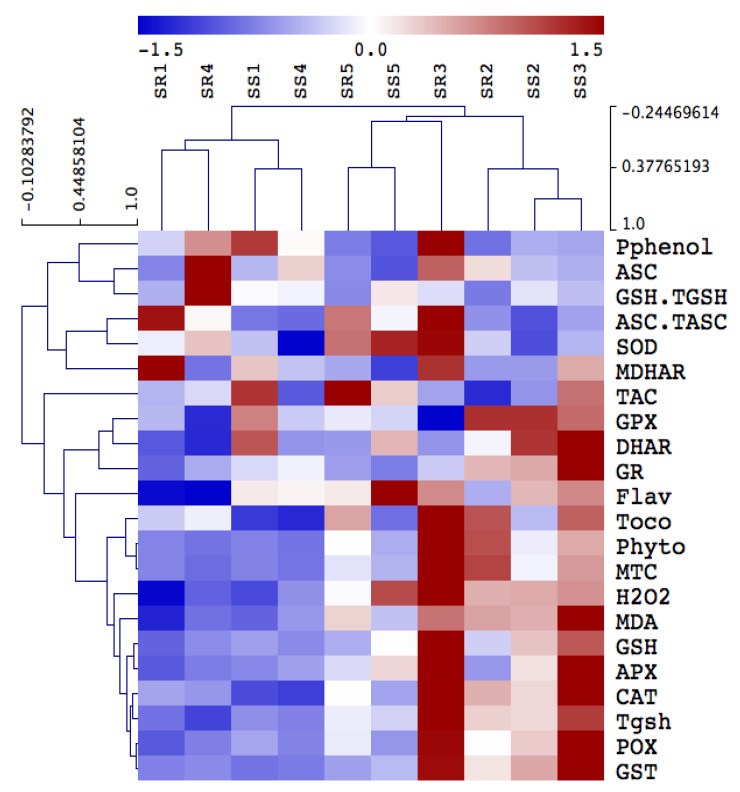
Hierarchical clustering analysis of the impact of soil pollution on oxidative antioxidant defense system in shoot and roots of *A. retroflexus*. The changes in antioxidants in *A. retroflexus* grown in control site (Site 5) and contaminated sites (Sites 1–4). Means of antioxidant metabolites and enzymes are hierarchically clustered and mean-centered. Ascorbate peroxidase (APX), monodehydroascorbate reductase (MDHAR), dehydroascorbate reductase (DHAR), glutathione reductase (GR), glutathione transferase (GST), peroxidase (GPX), superoxide dismutase (SOD), catalase (CAT), peroxidase (POX), FRAP (total antioxidant capacity), ascorbate (ASC), ascorbate redox status (ASC.TASC), reduced glutathione (GSH), total glutathione (Tgsh), glutathione redox status (GSH.TGSH) and tocopherols (Toco), hydrogen peroxide (H_2_O_2_), polyphenols (Pphenol), flavonoids (Flav), metallothioneins (MTC), phytochelatins (Phyto).

**Figure 5 biology-11-00164-f005:**
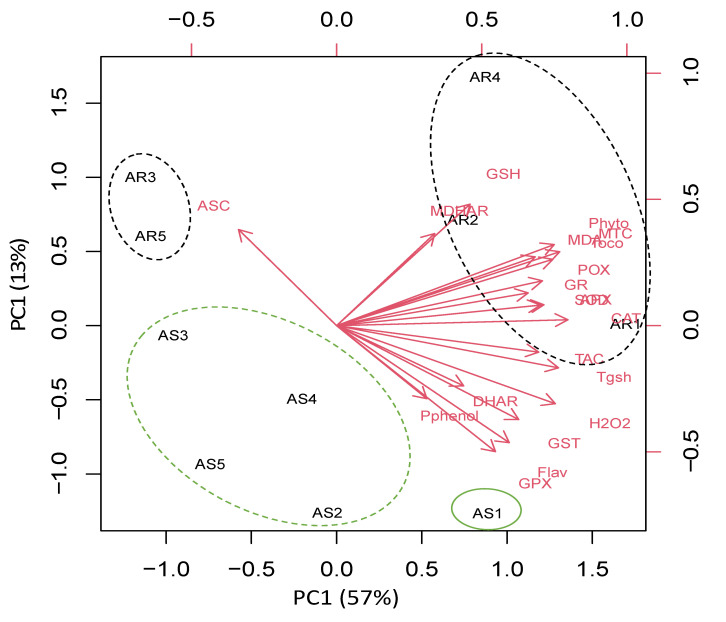
Principal component analysis (PCA) of parameters involved in photosynthesis, photorespiration, oxidative stress, detoxification, and antioxidant defense in roots and shoot of *Amaranthus retroflexus* grown in control site (Site 5) and contaminated sites (Sites 1–4). AR1-5 and AS1-5 means roots and shoot of *A. retroflexus*, respectively.

**Table 1 biology-11-00164-t001:** Monthly variations of rainfall, relative humidity in the study area. The data were extracted from the Jeddah meteorological station, Saudi Arabia. Mx: maximum temperature; Mn: minimum temperature; M: mean temperature.

	Temperature °C			Relative Humidity (%)			Precipitation (mL)
	Mx	Mn	M	Mx	Mn	M	
January	26.8	18.2	22.3	81	8	46	0
February	28	19.6	23.6	87	14	53	1
March	30.5	20.7	25.3	95	9	55	6.2
April	32.8	23.2	27.8	83	17	53	0
May	35.9	26.8	31.2	84	8	49	0
June	37.9	27.3	32.3	86	5	52	0
July	38.5	29.1	33.6	93	12	49	2
August	38.7	30	34.2	81	19	52	0
September	38.7	30	33.9	88	5	58	TRACE
October	39.2	27	32.4	89	3	49	3.8
November	33.1	25.3	28.8	88	23	57	19
December	31.9	22.4	26.6	84	8	53	8

**Table 2 biology-11-00164-t002:** Biodiversity indices for the studied sites and the relative density, frequency, fresh weight (FW) and dry weight (DW) of *A. retroflexus*. Data of FW and DW are mean values ± SE (*n* = 10). Different letters indicate statistically significant difference between means of the same plant species at significance level at least (*p* ≤ 0.05).

	Site 1	Site 2	Site 3	Site 4	Site 5
Evenness (R)	0.65	0.54	0.41	0.43	0.45
Shannon Index	0.1292	0.1714	0.189	0.21	0.25
Species richness	12	21	22	23	34
Cover %	7	10	12	13	15
*A. retroflexus* relative density	16.7	15.3	6.7	2.6	0.52
*A. retroflexus* frequency	100	95	36	36	25
*A. retroflexus* FW (g/individual)	178 ± 2.3 c	189 ± 1.8 c	210 ± 2.4 b	254 ± 2.1 a	256 ± 3.1 a
*A. retroflexus* DW (g/individual)	53.4 ± 1.1 c	57. ± 1.02 c	63 ± 1.1 b	74.3 ± 1.6 a	76 ± 1.6 a

**Table 3 biology-11-00164-t003:** Concentration of heavy metals, minerals, phenols, acetic acid, pH, organic matter, electric conductivity and texture in the studied sites soils. Data are mean values ± SE (*n* = 4). Different letters (in the same row) indicate statistically significant difference between means for the same plant species at significance level at least (*p* ≤ 0.05).

	Site 1	Site 2	Site 3	Site 4	Site 5
Cd (mg/kg)	1.23 ± 0.71 a	1.07 ± 0.55 b	0.76 ± 0.22 b	0.19 ± 0.09 c	0.24 ± 0.17 c
Ni (mg/kg)	5.16 ± 0.37 a	4.90 ± 0.46 a	3.02 ± 0.62 b	1.16 ± 0.20 c	0.36 ± 0.01d
As (mg/kg)	5.65 ± 0.99 a	4.30 ± 0.68 a	2.02 ± 0.43 b	0.66 ± 0.22 c	0.02 ± 0.01d
Cu (mg/kg)	30.3 ± 6.44 a	19.35 ± 2.92 b	7.63 ± 0.32 c	1.26 ± 0.32d	0.03 ± 0.01e
Pb (mg/kg)	0.41 ± 0.07 a	0.28 ± 0.08 b	0.20 ± 0.08 b	0.18 ± 0.08 b	0.02 ±0.00 c
Co (mg/kg)	0.33 ± 0.13 a	0.31 ± 0.11 a	0.26 ± 0.07 a	0.30 ± 0.10 a	0.01 ± 0.01 b
Hg (mg/kg)	0.39 ± 0.14 a	0.34 ± 0.14 a	0.27 ± 0.15 a	0.29 ± 0.13 a	0.05 ± 0.01 b
Al (mg/kg)	0.65 ± 0.29 a	0.48 ± 0.16 a	0.63 ± 0.33 a	0.46 ± 0.19 a	0.08 ± 0.03 b
V (mg/kg)	0.82 ± 0.07 a	0.54 ± 0.17 b	0.47 ± 0.15 b	0.25 ± 0.07 c	0.04 ± 0.01d
Cr (mg/kg)	0.82 ± 0.42 a	0.58 ± 0.38 b	0.49 ± 0.29 b	0.25 ± 0.04 c	0.03 ± 0.02d
Zn (mg/kg)	0.88 ± 0.08 a	0.77 ± 0.07 a	0.59 ± 0.06 b	0.09 ± 0.01 c	0.04 ± 0.00d
Mn (mg/kg)	0.02 ± 0.00 c	0.036 ± 0.0 a b	0.04 ± 0.00 b	0.10 ± 0.01 a	0.03 ± 0.01 b
Mg (mg/kg)	0.02 ± 0.01 c	0.05 ± 0.02 b	0.02 ± 0.01 c	0.05 ± 0.01 b	0.09 ± 0.05 a
Ca (mg/kg)	0.04 ± 0.01 b	0.09 ± 0.02 a	0.04 ± 0.01 b	0.08 ± 0.02 a	0.04 ± 0.01 b
Ba (mg/kg)	0.03 ± 0.01 b	0.08 ± 0.02 a	0.03 ± 0.01 b	0.07 ± 0.02 a	0.03 ± 0.00 b
Fe (mg/kg)	0.02 ± 0.01 c	0.06 ± 0.02 b	0.02 ± 0.01 c	0.06 ± 0.02 b	0.53 ± 0.09 a
K (mg/kg)	0.84 ± 0.01 a	1.03 ± 0.03 a	0.84 ± 0.12 a	1.00 ± 0.01 a	0.89 ± 0.06 a
N (mg/kg)	7.11 ± 0.45 a	7.57 ± 0.76 a	7.22 ± 0.94 a	7.82 ± 1.17 a	7.59 ± 0.53 a
Phenols (mg/g)	49.42 ± 7.5 a	46.76 ± 4.7 a	17.94 ± 3.1 b	13.17 ± 1.0 b	13.84 ± 0.1 b
Citric acid (mg/g)	25.1 ± 0.28 a	8.41 ± 0.69 b	8.26 ± 1.73 b	4.47 ± 1.17 c	4.01 ± 0.88 c
pH	7.80 ± 0.09 a	7.5 ± 0.02 a	7.4 ± 0.03 a	7.3 ± 0.01 a	7.3 ± 0.01 a
O.M (%)	1.85 ± 0.03 a	1.43 ± 0.02 a	1.35 ± 0.02 a	1.40 ± 0.03 a	0.99 ± 0.09 b
E.C (ds m^−1^)	1.01 ± 0.01 a	0.97± 0.01 a	0.81 ± 0.01 b	0.88 ± 0.01 b	0.75 ± 0.01 b
Sands (%)	61.1 ± 1.3 a	67.74 ± 3.20 a	61.49 ± 1.60 a	70.78 ± 1.36 a	63.99 ± 0.04 a
Silts (%)	21.6 ± 0.89 a	17.90 ± 1.04 a	21.2 ± 0.89 a	16.00 ± 0.68 a	14.44 ±0.04 a
Clay (%)	17.31 ± 0.78 a	14.36 ± 1.05 a	17.31 ± 0.8/8 a	13.22 ± 0.74 a	11.57 ±0.7 a

**Table 4 biology-11-00164-t004:** Metal concentrations in the shoot and roots of *Amaranthus retroflexus* grown in control site (Site 5) and contaminated sites (Sites 1–4). Data are mean values ± SE (*n* = 4). Different letters indicate statistically significant difference between means of the same plant species at significance level of at least *p* ≤ 0.05.

	Shoot					Roots				
	Site 1	Site 2	Site 3	Site 4	Site 5	Site 1	Site 2	Site 3	Site 4	Site 5
Cd	23.24 ± 6.6 a	19.9 ± 5.5 a	11.9 ± 2.2 b	3.02 ± 0.7 c	1.1 ± 0.2 d	66.4 ± 12 a	57.1 ± 10 a	32 ± 3.3 b	10.30 ± 1.9 c	7 ±0.8 d
Ni	161.3 ± 30 a	160.2 ± 29 a	90 ± 1.8 b	32.7 ± 3.4 c	1.50 ± 0.1 d	486 ± 109 a	480 ± 118 a	256 ± 53 b	117.1 ± 31 c	16.9 ± 4.1 d
As	182.4 ± 36 a	138 ± 27 b	58.5 ± 5.8 c	18.2 ± 3.5 d	0.11 ± 0.03 e	539 ± 113 a	411 ± 88 a	162 ± 23 b	63.1 ± 11 c	1.24 ± 0.1 d
Cu	1021 ± 190 a	593 ± 113 b	216 ± 4.0 c	32. ± 5.1 d	0.14 ± 0.02 e	3081 ± 818 a	1767 ± 4 b	616± 15 c	113.9 ± 22 d	1.58 ± 0.2 e
Pb	5.07 ± 1.21 a	3.45 ± 0.75 a	2.3 ±0.27 b	1.9 ± 0.29 b	0.12 ± 0.01 c	15.7 ± 5.69 a	10.6 ± 3 b	6.69 ± 2.2 c	7.07 ± 2.2 c	0.62 ± 0.22 d
Co	3.95 ± 0.74 a	3.7 ± 0.70 a	2.7 ± 0.19 b	2.98 ± 0.35 b	0.06 ± 0.02 c	11.92 ± 3.2 a	11.2± 3 a	7.8 ± 2.3 b	10.75 ± 3 b	0.3 ± 0.04 c
Hg	4.85 ± 0.95 a	4.3 ± 0.88 a	3.12 ±0.08 b	3.2 ± 0.54 b	0.21± 0.12 c	14.7 ± 4.4 a	13.3 ± 4.1 a	8.8 ± 2.2 b	11.7 ± 3.9 a	0.93 ± 0.24 c
Al	8.05 ± 1.52 a	6.0 ± 1.30 a	6.12 ± 0.44 a	4.2 ± 0.4 b	0.31± 0.08 c	24.3 ± 6.7 a	18.4 ± 6 b	17.1 ± 2.8 b	15.2 ± 3.6 b	1.5 ± 0.25 c
V	9.83 ± 2.2 a	6.1± 1.16 a	4.7 ± 0.20 b	2.6 ± 0.40 b	0.18 ± 0.02 c	30.22 ± 10 a	18.6 ± 5 b	13.6 ± 3 c	9.45 ± 3 d	0.93 ± 0.26 e
Cr	10.73 ± 2 a	7.7 ± 1.51 ab	5.7 ± 0.47 b	2.09 ± 0.42 c	0.17 ± 0.03 d	32.82 ± 10 a	23.5 ± 6 b	16.6 ± 5 c	7.70 ± 2.2 d	0.88 ± 0.22 e
Zn	10.4 ± 2.9 a	9.1± 2.5 a	6.22 ± 1.48 b	0.95 ± 0.26 c	0.19 ± 0.02 d	39.64 ± 9 a	34.6 ± 8 a	22.3 ± 4 b	4.28 ± 1 c	1.24 ± 0.19 d
Mn	0.64 ± 0.07 c	1.01 ± 0.1 b	1.18 ± 0.01 b	2.7 ± 0.30 a	0.90 ± 0.14 b	0.93 ± 0.06 c	1.4 ± 0.1 b	1.60 ± 0.2 b	4.11 ± 0.2 a	1.29 ± 0.20 b
Fe	0.80 ± 0.13 c	1.88 ± 0.3 b	0.78 ± 0.09 c	1.69 ± 0.2 b	13.5 ± 1.3 a	1.17 ± 0.1 d	2.7 ± 0.3 b	1.053 ± 0.1 c	2.2 ± 0.6 b	19.35 ± 1.9 a
K	22 ± 2.2 a	27 ± 2.8 a	24.8 ± 0.89 a	26.9 ± 2.7 a	23.6 ± 2.5 a	32.2 ± 2 b	40.5 ± 2 a	33 ± 3.6 b	40.8 ± 0.24 a	33.75 ± 3 b
N	187.3 ± 19 b	200 ± 22 a	212 ± 5.5 a	209 ± 27 a	200 ± 2 a	273 ± 17 b	292 ± 23 b	286 ± 30.2 b	341. ± 47 a	286.8 ± 30 b
Mg	0.68 ± 0.08 c	1.6 ± 0.2 b	0.67 ± 0.02 c	1.44 ± 0.19 b	2.4 ± 0.6 a	0.99 ± 0.09 c	2.3 ± 0.2 b	0.91 ± 0.1 c	2.00 ± 0.34 b	3.53 ± 0.8 a
Ca	1.16 ± 0.17 b	2.7 ± 0.4 a	1.16 ± 0.10 b	2.41 ± 0.3 a	1.18 ± 0.1 b	1.69 ± 0.2 b	3 ± 0.48 a	1.57 ± 0.2 b	3.26 ± 0.77 a	1.7 ±0.19 b
Ba	0.97 ± 0.13 b	2.2 ± 0.3 a	0.95 ± 0.07 b	2.05 ± 0.29 a	0.9 ± 0.10 b	1.42 ± 0.15 b	3.3 ± 0.3 a	1.2 ± 0.17 b	2.81 ± 0.59 a	1.3 ± 0.14 b

**Table 5 biology-11-00164-t005:** Biological concentration factor (BCF) and translocation factor (TF) of *Amaranthus retroflexus* grown in control site (Site 5) and contaminated sites (Sites 1–4). Data are mean values ± SE (*n* = 4). Different letters indicate statistically significant difference between means of the same plant species at significance level of at least *p* ≤ 0.05.

	Site 1		Site 2		Site 3		Site 4			Site 5
	BCF	TF	BCF	TF	BCF	TF	BCF	TF	BCF	TF
Cd	53.9 ± 2 a	0.34 ± 0.0 a	53.4 ± 2 a	0.3 ± 0.0 a	42.8 ± 2.4 b	0.36 ± 0.0 a	54.2 ± 1.2 a	0.29 ± 0.0 a	30 ± 1.3 c	0.16 ± 0.01 b
Pb	94 ± 1.2 a	0.32 ± 0.0 a	97.9 ± 2 a	0.32 ± 0.0 a	84.9 ± 2.4 b	0.34 ± 0.3 a	101 ± 2.1 a	0.27 ± 0.4 ab	46 ± 1 c	0.19 ± 0.02 b
Ni	95 ± 11 a	0.3 ± 0.04 a	95.6 ± 1 a	0.33 ± 0.1 a	80.5 ± 1.4 b	0.35 ± 0.06 a	95.6 ± 13 a	0.27 ± 0.05 b	62 ± 2.3 c	0.08 ± 0.01 c
Co	101 ± 1 a	0.3 ± 0.0 a	91.3 ± 1 a	0.3 ± 0.0 a	80.8 ± 11 b	0.34 ± 0.08 a	90.3 ± 1.0 a	0.27 ± 0.04 b	53 ± 1.3 c	0.21 ± 0.03 b
As	38 ± 14 a	0.3 ± 0.0 a	37 ± 15 a	0.3 ± 0.0 a	33.4 ± 1 ab	0.35 ± 0.01 a	39 ± 14 ab	0.28 ± 0.0 b	31 ± 2.1 c	0.09 ± 0.01 c
Hg	36.1 ± 1 a	0.32 ± 0.0 a	36.1 ± 2 a	0.3 ± 0.02 a	30.1 ± 1.2 ab	0.35 ± 0.0 a	35.8 ± 1.3 a	0.27 ± 0.04 b	30 ± 0.9 b	0.23 ± 0.05 b
Cu	37 ± 13 a	0.3 ± 0.0 a	39.2 ± 12 a	0.3 ± 0.01 a	32.9 ± 1.0 a	0.35 ± 0.01 a	40.4 ± 9.5 a	0.28 ± 0.06 a	18.6 ± 2 b	0.09 ± 0.02 b
Al	37 ± 0.0 a	0.3 ± 0.0 a	38 ± 0.0 a	0.3 ± 0.01 a	27.1 ± 7.8 b	0.35 ± 0.05 a	33 ± 0.03 a	0.21 ± 0.02 b	19 ± 0.01 c	0.20 ± 0.01 b
V	36.8 ± 1 a	0.32 ± 0.08 a	34.4 ± 1 a	0.33 ± 0.1 a	29.1 ± 0.0 b	0.34 ± 0.01 a	37.8 ± 1.6 a	0.27 ± 0.04 a	23.2 ± 1.4 c	0.19 ± 0.01 b
Cr	40.0 ± 1 a	0.32 ± 0.07 a	40.5 ± 0.1 a	0.32 ± 0 a	33.8 ± 1.02 b	0.34 ± 0.03 a	30.8 ± 1.2 b	0.27 ± 0.03ab	29.3 ± 1.2 b	0.20 ± 0.01 b
Zn	45.0 ± 2 a	0.26 ± 0.0 a	44. ± 1.6 a	0.2 ± 0.0 a	37.8 ± 0.12 b	0.27 ± 0.09 a	47.5 ± 2.1 a	0.22 ± 0.02 b	31.0 ± 1.02 c	0.15 ± 0.01 c
Fe	58.5 ± 2 a	0.7 ± 0.0 a	45.8 ± 2.8 a	0.7 ± 0.08 a	52.0 ± 1.03 a	0.74 ± 0.0 a	37.5 ± 2.7 b	0.75 ± 0.11 a	36.5 ± 3.6 b	0.70 ± 0.09 a

## Data Availability

Data presented in this study are available on reasonable request.

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
