# Peer review of "Heavy Metal Effects on Biodiversity and Stress Responses of Plants Inhabiting Contaminated Soil in Khulais, Saudi Arabia"

_biology, 2022, doi:10.3390/biology11020164_

Round 1

Reviewer 1 Report

This work is very interesting. The only thing I am hesitant is the approach of the study. 

Methods

Line 137 Section 2.5

“Violante et al. [32] examined the quantities of Cd, Ni, As, Hg, Al, Cu, V, Co, Fe, Cr, Pb, and Zn in ground plant material digested in a strong acid mixture.”

This statement is not clear.

Study approach

The study concentrates heavily on the toxicological parameters of sewage-impacted on one plant species of which the degree of toxicity exposure is the location from the sewage area with the farther the location, the less toxic the assumption of exposure. I found this approach difficult to get a clear conclusion as plant variation in toxicological parameters can vary based on genetic variation and environmental variation and may not be toxic-location related. The author needs to justify this approach in the introduction and find similar works to discuss the result from this approach.

A clear experimental approach would be to obtain the homogenous source of seed or planting material of the target plant and grow them at various locations with the farthest as the baseline control.

Another approach is to transfer the polluted soils at various distances and grow the plant under controlled environment and toxicity parameters measured. Then this baseline data can be used to look at correlation with real life ecosystem.

Results

Table 3. Concentration of heavy metals, minerals, phenols, acetic acid, pH, organic matter, electric 266

conductivity and texture in the studied sites soils. Data are mean values ± SE (n=4). Different letters 267

indicate statistically significant difference between means for the same plant species at significance 268

level at least (P≤0.05).

Should mention different letters in the same row or column

Suggest

Table 3 and 4 converted to bar diagram for easier visual interpretation

Author Response

Responses to Reviewer 1

Comments and Suggestions for Authors

Reviewer comment: This work is very interesting. The only thing I am hesitant is the approach of the study. 

Response: Thanks for positive feedback, here we aimed to introduce a heavy metal resistant species and to assess its phytoremediation potentiality and stress defense strategies. Our approach to achieve this goal was to do field research to compare the vegetation of polluted sites to that of a healthy site. For selecting the study area and doing vegetation surveys, scientific methods are applied. More datils were added to methods section for clarity.

Methods

Line 137 Section 2.5

Reviewer comment: “Violante et al. [32] examined the quantities of Cd, Ni, As, Hg, Al, Cu, V, Co, Fe, Cr, Pb, and Zn in ground plant material digested in a strong acid mixture.” This statement is not clear.

Response: thanks, The statement has been corrected

 Study approach

 Reviewer comment: The study concentrates heavily on the toxicological parameters of sewage-impacted on one plant species of which the degree of toxicity exposure is the location from the sewage area with the farther the location, the less toxic the assumption of exposure. I found this approach difficult to get a clear conclusion as plant variation in toxicological parameters can vary based on genetic variation and environmental variation and may not be toxic-location related. The author needs to justify this approach in the introduction and find similar works to discuss the result from this approach.

Response: Thanks for valuable comment, the studied area is quite small, the distance between the first and last contaminated sites is 5 kilometers, thus there is no environmental variation between the contaminated sites. Soils structures were also similar, thus we relied on location from the sewage area because we found that the closer to the pollution center, the more contaminated the soil with heavy elements, and the further away from the pollution center, the lower the concentration of heavy elements. This was proven by soil analysis, which included in the results of the current research.

Furthermore, A. retroflexus plants collected from different sites and of the same age were screened for the same genotype by our university colleagues and plant taxonomists.

Similar studies were also introduced in introduction part.

Reviewer comment: A clear experimental approach would be to obtain the homogenous source of seed or planting material of the target plant and grow them at various locations with the farthest as the baseline control.

Response: thanks, we thank the reviewer for this comment, and I will take this into account in the future research. Here we also carefully checked the collected plant martials and we only harvested A. retroflexus plants from different sites of the same genotype as reported above.

Reviewer comment: Another approach is to transfer the polluted soils at various distances and grow the plant under controlled environment and toxicity parameters measured. Then this baseline data can be used to look at correlation with real life ecosystem.

Our response: We thank the reviewer for the valuable comment, we currently managed to get the seeds and lab experiment will be conducted to further investigated the tolerance of A. retroflexus to individual and combination of different heavy metals. Because we think that it will be difficult  

Results

Table 3. Concentration of heavy metals, minerals, phenols, acetic acid, pH, organic matter, electric

conductivity and texture in the studied sites soils. Data are mean values ± SE (n=4). Different letters

indicate statistically significant difference between means for the same plant species at significance

level at least (P≤0.05).

Reviewer comment: Should mention different letters in the same row or column 

Response: thanks, different letters in the same row.

Reviewer comment: Table 3 and 4 converted to bar diagram for easier visual interpretation

Response: thanks, we understand that the table is quite complex. We convert it to bar diagram, but we did find that it is make it more complicated thus we decided to keep it in tubular form. Moreover, we re-arrange the data to make it clearer.

Reviewer 2 Report

The manuscript entitled “Heavy metal effects on biodiversity and stress responses of plants inhabiting contaminated soil in Khulais, Saudi Arabia” investigated the biodiversity of natural vegetation in the polluted sites and the control site, and explored the effect of heavy metal pollution on tolerant plant species. They find the plant richness reduced by the sewage polluted, and a heavy metal tolerant species, Amaranthus retroflexus L. which showed high BCF and low TF for Cu, As, and Ni, and also mitigate heavy metal oxidative stress by antioxidants mechanism in root. Thus, the authors suggested the potentially of A. retroflexus as a promising bioremediator. The article is easy to read. However, there are still some issues should be addressed.

  1. Introduction section should be improved. What’s the relationship between sewage water pollution and heavy metal pollution? What’s the progress of studies on the impact of sewage pollution and heavy metal pollution on the natural vegetation? What’s the novelty of this study?
  2. Results section: the authors should show the heavy metal pollution degree of the sewage water and the studied soils firstly. Then, the effect of pollution on floristic composition.
  3. 2 What did the abbreviated words mean?
  4. Many errors in the chemical formula. Please check them.

Author Response

Responses to Reviewer 2

Comments and Suggestions for Authors

The manuscript entitled “Heavy metal effects on biodiversity and stress responses of plants inhabiting contaminated soil in Khulais, Saudi Arabia” investigated the biodiversity of natural vegetation in the polluted sites and the control site, and explored the effect of heavy metal pollution on tolerant plant species. They find the plant richness reduced by the sewage polluted, and a heavy metal tolerant species, Amaranthus retroflexus L. which showed high BCF and low TF for Cu, As, and Ni, and also mitigate heavy metal oxidative stress by antioxidants mechanism in root. Thus, the authors suggested the potentially of A. retroflexus as a promising bioremediator. The article is easy to read. However, there are still some issues should be addressed.

Reviewer comment: Introduction section should be improved. What’s the relationship between sewage water pollution and heavy metal pollution? What’s the progress of studies on the impact of sewage pollution and heavy metal pollution on the natural vegetation? What’s the novelty of this study?

Response: Thanks, a relation the relationship between sewage water pollution and heavy metal pollution is added to the introduction (Line………..)

The following section in the introductions shows: What’s the progress of studies on the impact of sewage pollution and heavy metal pollution on the natural vegetation?

Response: thanks, Thanks, the progress of instigating the impact of heavy metal pollution on the natural vegetation is added. For instance, the studies on the impact of heavy metal pollution on the natural vegetation, could contribute to the evolution of species that are both sensitive and tolerant in response to heavy metal pollution

Reviewer comment: The following section in the introductions shows: What’s the novelty of this study?

Response: thanks, More details on study novelty is added at the end of the introduction

Reviewer comment: Results section: the authors should show the heavy metal pollution degree of the sewage water and the studied soils firstly. Then, the effect of pollution on floristic composition.

Our response: In the results, we presented a picture of the plant community present in the polluted and unpolluted sites to clarify the difference between the study sites and to determine the places for taking soil samples for analysis based on the difference in vegetation present.

Reviewer comment: What did the abbreviated words mean?

Response: thanks, abbreviations explained

Reviewer comment: Many errors in the chemical formula. Please check them.

Response: thanks, all chemical formula checked

Reviewer 3 Report

REFEREE REPORT FOR THE MANUSCRIPT Biology - 15516, entitled “Heavy metal effects on biodiversity and stress responses of plants inhabiting contaminated soil in Khulais, Saudi Arabia ”

In my opinion the research is interesting and appropriated for the journal Biology. Hence, my recommendation is the acceptance after minor revision.

I have a number of comments and suggestions that could improve the manuscript, in detail described below.

Abstract:

line 14: maybe some words about polluted sites selected

Introduction part:

A very complete study well conducted with a well-explained context is done. The paper is composed of a large set of complementary data allowing to identify the bioremediation and stress defense strategies of several tolerant plant species grown on contaminated soil.

line 46: microorganisms : term more large and appropriate

line 48: no link between the fact that plants are eukaryotes and after in the sentence

at the end of intro: no hypothesis formulated to more deeply explain the study done

Mat & Meth:

The mat & meth part is very well described and precise.

Below some remarks and comments:

line 89: sites localization should be indicated in details on a map

line 114: "detected" not appropriate

line 133: "five samples..." of what : not clear !

line 145: paragraph 2.7 should be revised: 2 parts similar ?

line 148: ratio : v/v ?

line 151: milliQ not miliQ

line 156: if only citric acid analyzed: title without S for "acids"

line 165: the end of sentence missing !

line 171: "in in"

line 175: acronym FRAP should be defined !

line 176: details missing and explanations not clear !

line 186: "activity" should be deleted

line 190: extraction not estraction

Results section

The results are well presented and properly analyzed; the essential facts that have emerged are highlighted. The explanations provided are supported by references ; the originality of the work carried out comes in particular from the diversity of approaches chosen.

line 251: arsenic is not a metal but a metalloid; terms "heavy metals "should be not used !

Discussion part:

The discussion judiciously provides clear explanations for the results obtained; it is based on relevant and recent references.

Conclusion:

The conclusion provides a good summary of the highlights of this study, highlighting the strong results. However (line 557) perspectives should be more explained.

Tables & figures:

Table 1, line 98: legend not enough developed: significance of Mx, Mn et M ?

Fig. 1, line 88: scale should be added

Fig. 2, line 221: chorology: part (b) of the figure?

Fig. 2(a) & (b): all abbreviation used should be defined in the legend!

Table 4: "N", "K" are not a heavy metals; legend have to be modified

Fig.4: legend should be completed; some abbreviations have to be defined

Author Response

Responses to Reviewer 3

Comments and Suggestions for Authors

REFEREE REPORT FOR THE MANUSCRIPT Biology - 15516, entitled “Heavy metal effects on biodiversity and stress responses of plants inhabiting contaminated soil in Khulais, Saudi Arabia ”

 In my opinion the research is interesting and appropriated for the journal Biology. Hence, my recommendation is the acceptance after minor revision.

I have a number of comments and suggestions that could improve the manuscript, in detail described below.

Abstract:

Reviewer comment: line 14: maybe some words about polluted sites selected

Response: thanks, Thanks, the sentence has been changed

Introduction part:

A very complete study well conducted with a well-explained context is done. The paper is composed of a large set of complementary data allowing to identify the bioremediation and stress defense strategies of several tolerant plant species grown on contaminated soil.

Reviewer comment: line 46: microorganisms: term larger and more appropriate

Response: thanks, the term microorganisms added

Reviewer comment: line 48: no link between the fact that plants are eukaryotes and after in the sentence

Our response: linked by the word (so)

Reviewer comment: at the end of intro: no hypothesis formulated to more deeply explain the study done

 Response: thanks, Thanks, these details are added to the end of the introduction.

Mat & Meth:

The mat & meth part is very well described and precise.

Below some remarks and comments:

Reviewer comment: line 89: sites localization should be indicated in details on a map

Our response: Indeed there is sites localization map, but unfortunately we could not obtain a license from Google Earth

Reviewer comment: line 114: "detected" not appropriate

Response: thanks, change to calculated

Reviewer comment: line 133: "five samples..." of what: not clear !

Response: thanks, Five plant samples (A. retroflexus)

Reviewer comment: line 145: paragraph 2.7 should be revised: 2 parts similar?

Response: thanks, revised

Reviewer comment: line 148: ratio: v/v ?

Response: thanks, Yes v/v added

Reviewer comment: line 151: milliQ not miliQ

Response: thanks, Corrected

Reviewer comment: line 156: if only citric acid analyzed: title without S for "acids"

Response: thanks, Corrected

Reviewer comment: line 165: the end of sentence missing !

Response: thanks, Corrected

Reviewer comment: line 171: "in in"

Response: thanks, one “in” removed

Reviewer comment: line 175: acronym FRAP should be defined !

Response: thanks, added

Reviewer comment: line 176: details missing and explanations not clear !

Response: thanks, Thanks, corrcetd

Reviewer comment: line 186: "activity" should be deleted

Response: thanks, removed

Reviewer comment: line 190: extraction not estraction

Response: thanks, Corrected (extraction)

Results section

The results are well presented and properly analyzed; the essential facts that have emerged are highlighted. The explanations provided are supported by references ; the originality of the work carried out comes in particular from the diversity of approaches chosen.

Reviewer comment: line 251: arsenic is not a metal but a metalloid; terms "heavy metals "should be not used !

Response: thanks, Corrected

Discussion part:

The discussion judiciously provides clear explanations for the results obtained; it is based on relevant and recent references.

Conclusion:

The conclusion provides a good summary of the highlights of this study, highlighting the strong results.

Reviewer comment: However (line 557) perspectives should be more explained.

Response: thanks, Thanks, more perspectives is added

 However, further investigations are needed to unfold these observed differences, and to identify molecular mechanism underlying heavy metals stress responses in root and shoot organs in plants.

Tables & figures:

Reviewer comment: Table 1, line 98: legend not enough developed: significance of Mx, Mn et M ?

Response: thanks, Abbreviation added

Reviewer comment: Fig. 1, line 88: scale should be added

Response: thanks, Scale Added

Reviewer comment: Fig. 2, line 221: chorology: part (b) of the figure?

Response: thanks, part (b) Added

Reviewer comment: Fig. 2(a) & (b): all abbreviation used should be defined in the legend!

Response: thanks, all abbreviation added

Reviewer comment: Table 4: "N", "K" are not a heavy metals; legend have to be modified

Response: thanks, legend modified

Reviewer comment: Fig.4: legend should be completed; some abbreviations have to be defined

Response: thanks, abbreviations are idendfied.

Round 2

Reviewer 1 Report

I suggest the authors look back at all of the statistics in this paper and consult a statistician on assigning non-significant letter non-significant letter which is visually problematic in Fig. 3 (attached) and may exist in tables. The discussion then should follow correct assignment of letters.

Author Response

We reviewed all the statistics in this paper and consulted a fellow statistician and asked him to review them statistically about the assignment of the non-significant letter in figures and tables (changes are marked with yellow color), hence the discussion part was revised